# META-LEARNING BY HALLUCINATING USEFUL EXAMPLES

## ABSTRACT

Learning to hallucinate additional examples has recently been shown as a promising direction to address few-shot learning tasks, which aim to learn novel concepts from very few examples. The hallucination process, however, is still far from generating effective samples for learning. In this work, we investigate two important requirements for the hallucinator — (i) *precision*: the generated examples should lead to good classifier performance, and (ii) *collaboration*: both the hallucinator and the classification component need to be trained jointly. By integrating these requirements as novel loss functions into a general meta-learning with hallucination framework, our *model-agnostic* PrecisE Collaborative hAlluciNator (PECAN) facilitates data hallucination to improve the performance of new classification tasks. Extensive experiments demonstrate state-of-the-art performance on competitive *mini*ImageNet and ImageNet based few-shot benchmarks in various scenarios.

## 1 INTRODUCTION

Modern deep learning models rely heavily on large amounts of annotated examples (Deng et al., 2009). Their data-hungry nature limits their applicability to real-world scenarios, where the cost of annotating examples is prohibitive, or they involve rare concepts (Zhu et al., 2014; Fink, 2011). In contrast, humans can grasp a new concept rapidly and make meaningful generalizations, even from a single example (Schmidt, 2009). To bridge this gap, there has been a recent resurgence of interest in *few-shot learning* that aims to learn novel concepts from very few labeled examples (Fei-Fei et al., 2006; Vinyals et al., 2016; Wang & Hebert, 2016; Snell et al., 2017; Finn et al., 2017).

Existing work tries to solve this problem from the perspective of meta-learning (Thrun, 1998; Schmidhuber, 1987), which is motivated by the human ability to leverage prior *experiences* when tackling a new task. Unlike the standard machine learning paradigm, where a model is trained on a set of exemplars, meta-learning is performed on a set of tasks, each consisting of its own training and test sets (Vinyals et al., 2016). By sampling small training and test sets from a large collection of labeled examples of *base* classes, meta-learning based few-shot classification approaches learn to extract task-agnostic knowledge, and apply it to a new few-shot learning task of *novel* classes.

One notable type of task-agnostic (or meta) knowledge comes from the shared mechanism of data augmentation or *hallucination* across categories (Wang et al., 2018; Gao et al., 2018; Schwartz et al., 2018; Zhang et al., 2018a). Hallucinating additional training data by generating images may seem like an easy solution for few-shot learning, but it is often challenging. In fact, the success of this paradigm is usually restricted to certain domains like handwritten characters (Lake et al., 2013), or requires additional supervision (Dixit et al., 2017; Zhang et al., 2018b) or sophisticated heuristics (Hariharan & Girshick, 2017). An alternative to generating raw data in the form of visually realistic images is to hallucinate examples in a learned feature space (Wang et al., 2018; Gao et al., 2018; Schwartz et al., 2018; Zhang et al., 2018a; Xian et al., 2019). This can be achieved by, for example, integrating a "hallucinator" module into a meta-learning framework, where it generates hallucinated examples, guided by real examples (Wang et al., 2018). The learner then uses an augmented training set which includes both the real and the hallucinated examples to learn classifiers. While the existing approaches showed that it is possible to adjust the hallucinator to generate examples that are helpful for classification, the generation process is *still far from producing effective samples* in the few-shot regime. Our key insight is that, to facilitate data hallucination to improve the performance of new classification tasks, two important requirements should be satisfied: (i) *precision*: the generated

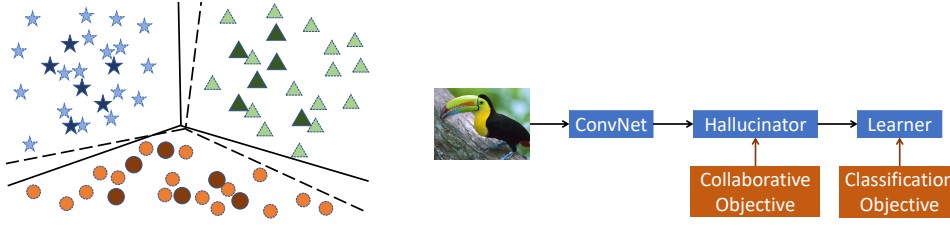

(a) *Precise hallucinator*      (b) *Collaborative hallucinator*

Figure 1: Illustration of the two important properties of our precise collaborative hallucinator, which facilitate data hallucination to improve the performance of classification tasks. (a) Precision: a classifier trained on hallucinated examples should match the performance of a classifier trained on real examples, demonstrated by the closeness of their decision boundaries. Real examples and their classifier are shown as dark colored shapes and solid lines, respectively; hallucinated examples and their classifier are shown as light colored shapes and dashed lines, respectively. (b) Collaboration: all the components need to be trained jointly. In addition to the classification objective imposed on the learner, a collaborative objective is introduced on the hallucinator as direct and early supervision. We integrate these properties into the meta-learning with hallucination framework for few-shot learning.

examples should lead to good classifier performance, and (ii) *collaboration*: all the components including the hallucinator and the learner need to be trained jointly.

In this work, we propose *PrecisE Collaborative hAlluciNator (PECAN)*, which integrates these requirements into a general meta-learning with hallucination framework, as shown in Figure 1. Assume that we have a hallucinator to generate additional examples from the original *small* training set. A precise hallucinator indicates that a classifier trained on both the hallucinated and the few real examples should produce superior validation accuracy. This can be achieved by training the hallucinator end-to-end with the learner, and back-propagating a classification loss based on ground-truth labels of validation data (Wang et al., 2018). Since this precision is measured using ground-truth labels, we term it as *hard precision*. And more importantly, if the hallucinator perfectly captures the target distribution, a classifier trained on a set of hallucinated examples, despite being generated from a small set of real examples, should produce roughly the same validation accuracy as a classifier trained on a large set of real examples, when these two sets are of the same sample size (Shmelkov et al., 2018). This indicates similar level of realism and diversity between the generated and the real examples, as shown in Figure 1a. Motivated by this observation, we introduce an additional *precision-inducing* loss function, which explicitly encourages the hallucinator to generate examples so that a classifier trained on them makes predictions similar to the one trained on a large amount of real examples. Given that this precision is measured based on classifier predictions, we term it as *soft precision*. This precision, which is complementary to hard precision and effective, as shown in our experiment, is lacking in current approaches (Wang et al., 2018).

Satisfying the precision requirement alone is not sufficient, since the classification objective is still directly associated with the learner, and thus the hallucinator continues to rely on the back-propagated signal to update its parameters. This leads to a potential undesirable effect of *imbalanced training* between the hallucinator and the learner: the learner tends to be stronger and makes allowances for errors in the hallucination, whereas the hallucinator becomes "lazy" and does not make its best effort to capture the data distributions, which is empirically observed in our experiments (See Figure 3). To address this issue, our key insight is to enforce *direct and early supervision* for the hallucinator, and make its contribution to the overall classification transparent, as shown in Figure 1b. Hence, we introduce a *collaborative objective* for the hallucinator, which allows us to directly influence the generation process to favor highly discriminative examples *right after hallucination*, and to strengthen the cooperation between the hallucinator and the learner.

**Our contributions** are three-fold. (1) We propose a novel loss that helps produce precise hallucinated examples, by using the classifier trained on real examples as a guidance, and encouraging the classifier trained on hallucinated examples to mimic its behavior. (2) We introduce a collaborative objective for the hallucinator as early supervision, which directly facilitates the generation process and improves the cooperation between the hallucinator and the learner. (3) By integrating these properties, we develop a general meta-learning with hallucination framework, which is *model-agnostic* and can be combined with any meta-learning models to consistently boost their few-shot learning performance.

Here we mainly focus on few-shot classification tasks, and we show that our approach applies to few-shot regression tasks as well in the appendix A.7.

## 2 RELATED WORK

As one of the unsolved problems in machine learning and computer vision, few-shot learning is attracting growing interest in the deep learning era (Miller et al., 2000; Fei-Fei et al., 2006; Lake et al., 2015; Santoro et al., 2016; Wang & Hebert, 2016; Vinyals et al., 2016; Snell et al., 2017; Finn et al., 2017; Hariharan & Girshick, 2017; George et al., 2017; Triantafillou et al., 2017; Edwards & Storkey, 2017; Mishra et al., 2018; Douze et al., 2018; Wang et al., 2018; Chen et al., 2019a; Dvornik et al., 2019). Successful generalization from few training samples requires appropriate "inductive biases" or shared knowledge from related tasks (Baxter, 1997), which is commonly acquired through transfer learning and more recently meta-learning (Thrun, 1998; Schmidhuber, 1987; Schmidhuber et al., 1997; Bengio et al., 1992). By explicitly "learning-to-learn" over a series of few-shot learning tasks (i.e., episodes), which are simulated from base classes, meta-learning exploits accumulated task-agnostic knowledge to target few-shot learning problems of novel classes. Within this paradigm of approaches, various types of meta-knowledge has been recently explored, including (1) a generic feature embedding or metric space, in which images are easy to classify using a distance-based classifier such as cosine similarity or nearest neighbor (Koch et al., 2015; Vinyals et al., 2016; Snell et al., 2017; Sung et al., 2018; Ren et al., 2018; Oreshkin et al., 2018); (2) a common initialization of network parameters (Finn et al., 2017; Nichol & Schulman, 2018; Finn et al., 2018) or learned update rules (Andrychowicz et al., 2016; Ravi & Larochelle, 2017; Munkhdalai & Yu, 2017; Li et al., 2017; Rusu et al., 2019); (3) a transferable strategy to estimate model parameters based on few novel class examples (Bertinetto et al., 2016; Qiao et al., 2018; Qi et al., 2018; Gidaris & Komodakis, 2018), or from an initial small dataset model (Wang & Hebert, 2016; Wang et al., 2017).

Complementary to these discriminative approaches, our work focuses on synthesizing samples to deal with data scarcity. There has been progress in this direction of data hallucination, either in pixel or feature spaces (Salakhutdinov et al., 2012; George et al., 2017; Lake et al., 2013; 2015; Wong & Yuille, 2015; Rezende et al., 2014; Goodfellow et al., 2014; Radford et al., 2016; Dixit et al., 2017; Hariharan & Girshick, 2017; Wang et al., 2018; Gao et al., 2018; Schwartz et al., 2018; Zhang et al., 2018a). However, it is still challenging for modern generative models to capture the entirety of data distribution (Salimans et al., 2016) and produce useful examples that maximally boost the recognition performance (Wang et al., 2018), especially in the small sample-size regime. In the context of generative adversarial networks (GANs), Shmelkov et al. (2018) show that images synthesized by state-of-the-art approaches, despite their impressive visual quality, are insufficient to tackle recognition tasks, and encourage the use of quantitative measures based on classification results to evaluate GAN models. Rather than using classification results as a performance measure, we go a step further in this paper by leveraging classification objectives to guide the generation process.

Other related work such as Wang et al. (2018) proposed a general data hallucination framework based on meta-learning, which is a special case of our approach. A GAN-like hallucinator takes a seed example and a random noise vector as input to generate a new sample. This hallucinator is trained jointly with the classifier in an end-to-end manner. Delta-encoder (Schwartz et al., 2018) is a variant of Wang et al. (2018), where instead of using noise vectors, it modifies an auto-encoder to extract transferable intra-class deformations, i.e., "deltas", and applies them to novel samples to generate new instances. Unlike the above approaches that directly use the produced samples to train the classifier, MetaGAN (Zhang et al., 2018a) trains the classifier in an adversarial manner to augment the classifier with the ability to discriminate between real and synthesized data. Another variant (Gao et al., 2018) explicitly preserves covariance information to enable better augmentation. Our work investigates critical yet unexplored properties in this paradigm that the data hallucinator should satisfy. These properties are general and can be flexibly incorporated into existing meta-learning approaches and hallucination methods, providing significant gains irrespective of these choices.

## 3 META-LEARNING WITH HALLUCINATION

We begin by presenting the general meta-learning mechanism (Vinyals et al., 2016; Snell et al., 2017; Finn et al., 2017) and our meta-learning with hallucination framework for the task of

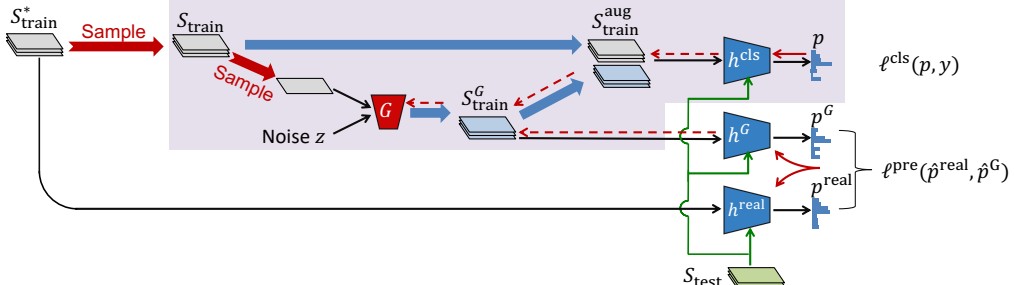

Figure 2: Meta-learning with our precise collaborative hallucinator. In each episode, given an initial (sampled) training set $S_{\text{train}}^*$, we sample its subset $S_{\text{train}}$. With real seed examples sampled from $S_{\text{train}}$ and noise vector $z$, we obtain a set of hallucinated examples $S_{\text{train}}^G$ through the generator $G$. $S_{\text{train}}$ and $S_{\text{train}}^G$ are combined to create an augmented training set $S_{\text{train}}^{\text{aug}}$. Conditioning on $S_{\text{train}}^{\text{aug}}$ ($S_{\text{train}}^G$, or $S_{\text{train}}^*$), a learner classification network $h^{\text{cls}}$ ($h^G$, or $h^{\text{real}}$) learns a new embedding space and outputs class probabilities $p$ ($p^G$, or $p^{\text{real}}$) for a set of real test examples $S_{\text{test}}$. The *classification objective* $\mathcal{L}_{\text{learner}}$ is a combination of the *hard precision* $\ell_{\text{learner}}^{\text{cls}}$ (classification loss calculated based on $p$ and ground-truth labels $y$) and the *soft precision-inducing* loss $\ell_{\text{learner}}^{\text{pre}}$ (calculated based on $\widehat{p}^{\text{real}}$ and $\widehat{p}^G$). The *collaborative objective* $\mathcal{L}_{\text{hal}}$ shares the same formulation of $\mathcal{L}_{\text{learner}}$ (i.e., a combination of $\ell_{\text{hal}}^{\text{cls}}$ and $\ell_{\text{hal}}^{\text{pre}}$), but is directly enforced *before the embedding layers* as early supervision for the hallucinator. The hallucinator and the learner are trained end-to-end based on the combination of $\mathcal{L}_{\text{learner}}$ and $\mathcal{L}_{\text{hal}}$. Dotted red arrows indicate the flow of gradients during back-propagation.

few-shot image classification. Let $\mathcal{I}$ be the space of images. We are given two *disjoint* sets of classes: a base class set $\mathcal{C}_{\text{base}}$ and an unseen novel class set $\mathcal{C}_{\text{novel}}$. The corresponding base dataset $D_{\text{base}} = \{(\mathbf{I}_i, y_i), \mathbf{I}_i \in \mathcal{I}, y_i \in \mathcal{C}_{\text{base}}\}$ contains a large number of labeled examples per class, while the novel dataset $D_{\text{novel}} = \{(\mathbf{I}_i, y_i), \mathbf{I}_i \in \mathcal{I}, y_i \in \mathcal{C}_{\text{novel}}\}$ consists of only a small number $n$ of labeled examples per class. The goal is to learn a classifier $h_{\theta_h}^{\text{cls}}$ parametrized by $\theta_h$ on $D_{\text{base}}$ that can cross-generalize (Bart & Ullman, 2005) to $\mathcal{C}_{\text{novel}}$ even when $n$ is as few as one.

Meta-learning aims to achieve such generalization through *episodic meta-training* that explicitly mimics the few-shot learning scenario on $D_{\text{base}}$ (Vinyals et al., 2016). Specifically, in each episode of the meta-training stage, the meta-learner simulates a few-shot classification task out of $D_{\text{base}}$. This task is constructed by first randomly sampling a subset of $m$ classes from $\mathcal{C}_{\text{base}}$, and then randomly sampling a small "training" set $S_{\text{train}}$ (also called the support set) and a small "test" set $S_{\text{test}}$ (also called the query set). The learner, i.e., the classifier $h_{\theta_h}^{\text{cls}}$, outputs estimated conditional probabilities $p$ for each example $(x, y)$ in $S_{\text{test}}$ based on $S_{\text{train}}$. That is, $p(x) = h_{\theta_h}^{\text{cls}}(x, S_{\text{train}})$. The meta-learner back-propagates the gradient of the total classification loss $\ell^{\text{cls}} = \sum_{(x,y) \in S_{\text{test}}} \text{loss}(h_{\theta_h}^{\text{cls}}(x, S_{\text{train}}), y)$ in $S_{\text{test}}$ to update the learner parameters $\theta_h$. During the *meta-testing* stage, the resulting $h_{\theta_h}^{\text{cls}}$ is used to address the few-shot classification task on $D_{\text{novel}}$, which predicts class probabilities of unlabeled test examples conditioned on the given small labeled training set $S_{\text{train}}$ of $\mathcal{C}_{\text{novel}}$.

Our meta-learning with hallucination framework introduces an additional "hallucinator" module $G_{\theta_G}$ with parameters $\theta_G$ to augment the small training set $S_{\text{train}}$. To facilitate training, we follow recent work (Hariharan & Girshick, 2017; Wang et al., 2018) and first pre-train a deep convolutional network on $D_{\text{base}}$ using a standard cross-entropy loss. We use it to extract the feature representation $x \in \mathcal{X}$ for an input image $\mathbf{I}$. Meta-learning is then performed over the pre-trained features $\{x_i\}$. As shown in the shaded region in Figure 2, given an initial $S_{\text{train}}$, the hallucinator $G_{\theta_G}$ generates additional examples for each class. Our framework applies to various types of hallucinators, and here we consider a powerful GAN-like hallucinator in Wang et al. (2018). Each hallucinated example is of the form $(G_{\theta_G}(x, z), y)$, where $(x, y)$ is a sampled seed example from $S_{\text{train}}$, and $z$ is a sampled noise vector. The set of generated examples $S_{\text{train}}^G$ is added to $S_{\text{train}}$ to create an augmented training set $S_{\text{train}}^{\text{aug}}$. In the next section, we show how to meta-train $G_{\theta_G}$ on $\mathcal{C}_{\text{base}}$, so that it can hallucinate new examples to augment $S_{\text{train}}$ of $\mathcal{C}_{\text{novel}}$ during meta-testing.

## 4  PRECISE COLLABORATIVE HALLUCINATOR

We now present our PrecisE Collaborative hAlluciNator (PECAN) shown in Figure 2, which exploits two important criteria for useful hallucination: *precision* and *collaboration*. As important constraints and guidance, these criteria facilitate hallucination to improve the classification performance.

**Basic hallucinator with hard precision.** At first, a precise hallucinator indicates that a classifier trained on $S_{\text{train}}^{\text{aug}}$ should produce superior validation accuracy. We achieve this by training the hallucinator end-to-end with the learner (Wang et al., 2018). As shown in the shaded region in Figure 2, during each episode of meta-training, the learner module $h_{\theta_h}^{\text{cls}}$ uses $S_{\text{train}}^{\text{aug}}$ to produce conditional probabilities $h_{\theta_h}^{\text{cls}}(x, S_{\text{train}}^{\text{aug}})$ for each example $(x, y)$ in the test set $S_{\text{test}}$. The meta-learner then back-propagates the gradient of the total classification loss $\ell^{\text{cls}} = \sum_{(x,y) \in S_{\text{test}}} \text{loss}(h_{\theta_h}^{\text{cls}}(x, S_{\text{train}}^{\text{aug}}), y)$ to update both the learner parameters $\theta_h$ and the hallucinator parameters $\theta_G$.

**Soft precision-inducing hallucinator.** One of the important characteristics of an optimal generative model is that the generated examples should be indistinguishable from real ones (Goodfellow et al., 2014). We argue that, in terms of our recognition task oriented hallucinator, this means that the classifier trained on hallucinated examples needs to be similar to the classifier trained on real examples. As shown in Figure 2, given an initial relatively large training set $S_{\text{train}}^{*}$, which contains $n^*$ examples for each of the $m$ classes, we randomly sample $n$ ($n \ll n^*$) examples per class, and obtain a subset $S_{\text{train}}$. From $S_{\text{train}}$, the hallucinator $G_{\theta_G}$ generates $n^*$ examples per class as $S_{\text{train}}^{G}$. This produces two training sets: $S_{\text{train}}^{*}$ with real examples and $S_{\text{train}}^{G}$ with hallucinated examples, where both contain the same number of examples. *Importantly*, note that $S_{\text{train}}^{G}$ is hallucinated from the subset $S_{\text{train}}$ instead of the initial large set $S_{\text{train}}^{*}$, and because $n \ll n^*$, we *rule out the trivial identity hallucinator or memorization*. We train two additional classification networks: $h^{\text{real}}$ based on $S_{\text{train}}^{*}$ and $h^{G}$ based on $S_{\text{train}}^{G}$, both of which have the same architecture as $h^{\text{cls}}$. When evaluated on the same test set $S_{\text{test}}$ composed of real examples, a comparable performance between $h^{\text{real}}$ and $h^{G}$ shows that the hallucinated samples are sufficiently precise, and as diverse as the real training set. Otherwise, when the hallucinator is imperfect, the accuracy of $h^{G}$ will be lower than that of $h^{\text{real}}$.

This similarity of classification accuracy essentially measures the difference between the learned (i.e., hallucinated) and the target (i.e., real) distributions, which could serve as an additional supervisory signal for training a better hallucinator. Since quantifying the similarity of accuracy directly would be difficult (Hinton et al., 2015), we instead introduce a loss function that acts on the network predictions. For an example $(x, y)$ in $S_{\text{test}}$, the two networks produce conditional probabilities

$$p^{\text{real}}(x) = h_{\theta_h}^{\text{real}}(x, S_{\text{train}}^{*}) \ \text{ and } \ p^{G}(x) = h_{\theta_h}^{G}(x, S_{\text{train}}^{G}), \tag{1}$$

respectively. While only the largest entry in $p^{\text{real}}(x)$ or $p^{G}(x)$ is used to make predictions associated with the ground-truth label $y$, other entries still carry rich information about the recognition task and the network, as observed in (Hinton et al., 2015; Dvornik et al., 2019). We thus leverage the probabilities $\widehat{p}^{\text{real}}$ and $\widehat{p}^{G}$ in the absence of the ground-truth label and measure their similarity using the negative cosine distance:

$$\psi(\widehat{p}^{\text{real}}, \widehat{p}^{G}) = -\cos(\widehat{p}^{\text{real}}, \widehat{p}^{G}), \tag{2}$$

where $\widehat{p}^{\text{real}}$ and $\widehat{p}^{G}$ are obtained by removing the logit for $y$ in $p^{\text{real}}$ and $p^{G}$, and re-normalizing the remaining logits using softmax with a learnable temperature. We treat the classification networks $h^{\text{cls}}$, $h^{\text{real}}$, and $h^{G}$ as the *new learner $h$* and use shared parameters for them. Their difference thus lies in different conditional training sets. We obtain the soft precision-inducing loss $\ell^{\text{pre}}$ by summing the loss (2) in $S_{\text{test}}$ and then combine it with the hard precision (i.e., the classification) loss as the *classification objective*. $p^{G}$ is now encouraged to not only make the right prediction according to the ground-truth label, but also make similar second-best, third-best, etc., choice predictions as $p^{\text{real}}$.

**Collaboration between hallucinator and learner.** We now consider the interaction between the hallucinator $G$ and the learner $h$. While hallucination is conducted in the pre-trained feature space $\mathcal{X}$, the final classification is performed in a new embedding space $\Phi$ learned by the learner. Since the classification objective is directly imposed on the learner $h$, the hallucinator $G$ continues to rely on the back-propagated signal to update its parameters. We may end up with a good embedding space $\Phi$ but a poor hallucinator $G$ in the original space $\mathcal{X}$. This undesired effect implies a potential *imbalance* between the hallucinator and the learner — *a stronger learner* that is able to make allowances for errors in hallucination, but a "lazy" hallucinator that does not make its best effort to capture the data distributions. Indeed, as is empirically validated in the experimental section (see Figure 3), despite being able to match the class distributions in the embedding space $\Phi$, the hallucinated examples are initially pulled away from the class distributions in the feature space $\mathcal{X}$.

To mitigate this issue, we introduce a simple *collaborative objective* to the hallucinator, which provides an additional constraint or regularization on the hallucination process. This collaborative

objective is the same as the above classification objective (i.e., a combination of the classification loss and the precision-inducing loss), but enforces *direct and early supervision* for the hallucinator in the pre-trained feature space $\mathcal{X}$. By doing so, we directly influence the update process of the hallucinator parameters, and generate much more discriminative examples *right after hallucination* than would be the case if we had to rely on gradual back-propagation from the learner alone. Our objective thus strengthens the cooperation between the hallucinator and the learner for the final classification performance, which can be viewed as a source of deep supervision that introduces auxiliary losses to intermediate layers when training deep neural networks (Simonyan & Zisserman, 2015; Lee et al., 2015). The overall objective combines the classification objective $\mathcal{L}_{\text{learner}}$ (on the learner) and the collaborative objective $\mathcal{L}_{\text{hal}}$ (on the hallucinator), each of which consists of a classification loss $\ell^{\text{cls}}$ (hard precision as cross-entropy with respect to ground-truth) and a soft precision-inducing loss $\ell^{\text{pre}}$:

$$\mathcal{L}(\theta_G, \theta_h) = \mathcal{L}_{\text{learner}} + \lambda \mathcal{L}_{\text{hal}} = \ell^{\text{cls}}_{\text{learner}} + \lambda_1 \ell^{\text{pre}}_{\text{learner}} + \lambda_2 \ell^{\text{cls}}_{\text{hal}} + \lambda_3 \ell^{\text{pre}}_{\text{hal}}, \tag{3}$$

where $\lambda$, $\lambda_1$, $\lambda_2$, and $\lambda_3$ are scalar hyper-parameters.

Our hallucinator is general and applies to different types of $h$ (i.e., meta-learning algorithms). Here we focus on the widely used and powerful prototypical networks (PN) (Snell et al., 2017), prototype matching networks (PMN) (Vinyals et al., 2016; Wang et al., 2018), and cosine classifiers (Cos) (Gidaris & Komodakis, 2018; Chen et al., 2019a). Without loss of generality, we take PN as an example to explain the overall meta-training and meta-testing process. PN learns an embedding space $\Phi$ and uses a non-parametric nearest centroid classifier to assign class probabilities for a test example based on its distances from class means in $\Phi$. As before, in each meta-training episode, after sampling $S^*_{\text{train}}$, $S_{\text{train}}$, and $S_{\text{test}}$ and hallucinating $S^G_{\text{train}}$ in the pre-trained feature space $\mathcal{X}$, we perform nearest centroid classification and produce the collaborative objective $\mathcal{L}_{\text{hal}}$ on $S_{\text{test}}$. We then feed the examples to the PN learner, obtain their embedded features in $\Phi$, perform nearest centroid classification, and produce the classification objective $\mathcal{L}_{\text{learner}}$ on $S_{\text{test}}$. The final loss is back-propagated to update both the PN learner parameters $\theta_h$ and the hallucinator parameters $\theta_G$. Figure 2 shows a schematic of the entire process. During meta-testing, we use the resulting $G_{\theta_G}$ to hallucinate new examples to augment $S_{\text{train}}$ of $\mathcal{C}_{\text{novel}}$, and we combine the predicted class probabilities in $\mathcal{X}$ and $\Phi$ as the final predictions.

## 5 EXPERIMENTAL EVALUATION

We explore the use of our meta-learning with hallucination framework for few-shot visual classification tasks. We focus the evaluation on the ImageNet based few-shot benchmark (Hariharan & Girshick, 2017; Wang et al., 2018). This is one of the largest datasets by far used for few-shot classification and it captures more realistic scenarios than others based on handwritten characters (Lake et al., 2015) or low-resolution images (Vinyals et al., 2016). The benchmark divides the 1,000 ImageNet categories (Russakovsky et al., 2015) into 389 base classes $\mathcal{C}_{\text{base}}$, with thousands of training images per class, and 611 novel classes $\mathcal{C}_{\text{novel}}$, with a small number $n$ of training images per class. Following Hariharan & Girshick (2017), we use $\mathcal{C}_{\text{base}}$ to train a convolutional network (ConvNet) based feature extractor and to conduct meta-training. Meta-testing is performed on $\mathcal{C}_{\text{novel}}$, and the performance is evaluated on a held-out test set, i.e., the original validation set of ImageNet. In addition, to avoid over-fitting, both $\mathcal{C}_{\text{base}}$ and $\mathcal{C}_{\text{novel}}$ are further split into two disjoint subsets. 193 of the base classes $\mathcal{C}^{\text{cv}}_{\text{base}}$ and 300 of the novel classes $\mathcal{C}^{\text{cv}}_{\text{novel}}$ are used for cross-validating hyper-parameters, and the remaining 196 base classes $\mathcal{C}^{\text{fin}}_{\text{base}}$ and 311 novel classes $\mathcal{C}^{\text{fin}}_{\text{novel}}$ are used for the final evaluation. Here we focus on hallucinating novel instances and thus evaluate the performance primarily on the novel classes $\mathcal{C}^{\text{fin}}_{\text{novel}}$, which is also consistent with most of the contemporary work (Vinyals et al., 2016; Snell et al., 2017; Finn et al., 2017). We report the mean top-1 and top-5 accuracies for 311-way, $n = 1, 2, 5, 10, 20$-shot classification, with each of them averaged over 5 trials.

In addition to this challenging version of ImageNet, we also evaluate on the widely used *mini*ImageNet (Vinyals et al., 2016) dataset to show the generality of our approach. *mini*ImageNet is a subset of 100 classes selected randomly from ImageNet with 600 images sampled from each class. Following the data split in Ravi & Larochelle (2017), we use 64 base, 16 validation, and 20 novel classes. We evaluate in the standard 5-way, 1-shot and 5-way, 5-shot settings (Vinyals et al., 2016).

| | Method (311-way classification) | Top-1 accuracy | | | | | Top-5 accuracy | | | | |
|---|---|---|---|---|---|---|---|---|---|---|---|
| | | n=1 | 2 | 5 | 10 | 20 | n=1 | 2 | 5 | 10 | 20 |
| Meta-learning method I | PMN w/ G + PECAN (Ours) | **21.3** | **29.0** | **39.1** | **45.3** | **49.6** | **47.0** | **59.1** | **70.5** | **75.5** | **78.7** |
| | PMN w/ G (Wang et al., 2018) | *21.0* | *28.4* | *38.2* | *43.9* | 47.9 | 45.8 | *57.8* | 69.0 | 74.3 | 77.4 |
| | PMN w/ aug | 20.3 | 27.7 | 37.9 | 43.7 | 47.5 | 44.1 | 56.4 | 68.8 | 74.1 | 77.2 |
| | PMN (Wang et al., 2018) | 19.6 | 27.2 | 37.5 | 43.5 | 47.3 | 43.3 | 55.7 | 68.4 | 74.0 | 77.0 |
| Meta-learning method II | PN w/ G + PECAN (Ours) | 20.8 | 27.7 | 37.5 | *43.9* | *48.4* | *46.5* | 57.5 | *69.3* | *74.9* | *78.1* |
| | PN w/ G (Wang et al., 2018) | 19.9 | 26.4 | 35.7 | 41.9 | 45.9 | 45.0 | 55.9 | 67.3 | 73.0 | 76.5 |
| | PN w/ aug | 18.7 | 25.9 | 34.9 | 39.6 | 42.3 | 40.2 | 55.0 | 66.7 | 71.6 | 74.3 |
| | PN (Snell et al., 2017) | 17.7 | 25.3 | 34.4 | 39.3 | 42.1 | 39.3 | 54.4 | 66.3 | 71.2 | 73.9 |
| Meta-learning method III | Cos-Cls w/ G + PECAN (Ours) | 18.8 | 25.4 | 33.7 | 38.1 | 39.2 | 43.1 | 53.6 | 64.8 | 69.9 | 71.4 |
| | Cos-Cls w/ G (Ours) | 18.4 | 25.1 | 33.5 | 37.5 | 38.6 | 42.1 | 52.8 | 64.0 | 69.0 | 70.7 |
| | Cos-Cls (Gidaris & Komodakis, 2018; Chen et al., 2019a) | 17.7 | 21.4 | 25.7 | 28.4 | 30.0 | 41.6 | 49.2 | 56.3 | 60.4 | 62.7 |
| Other Meta-learning Baselines | Cos & Att. (Gidaris & Komodakis, 2018) | - | - | - | - | - | 46.0 | 57.5 | 69.2 | 74.8 | *78.1* |
| | MN (Vinyals et al., 2016) | 19.8 | 25.8 | 34.8 | 41.1 | 46.5 | 43.6 | 54.0 | 66.0 | 72.5 | 76.9 |
| | MAML (Finn et al., 2017; Gao et al., 2018) | - | - | - | - | - | 39.2 | - | 64.2 | - | 76.8 |
| Non-meta-learning baselines | LogReg (Wang et al., 2018) | 16.8 | 24.9 | 35.6 | 42.2 | 48.0 | 38.4 | 51.1 | 64.8 | 71.6 | 76.6 |
| | LogReg w/ Analogies (Hariharan & Girshick, 2017) | 17.1 | 23.5 | 32.5 | 39.2 | 48.0 | 40.7 | 50.8 | 62.0 | 69.3 | 76.5 |
| | Gaussian hallucinator (Wang et al., 2018) | 16.7 | 24.2 | 33.4 | 38.2 | 44.0 | 39.1 | 51.4 | 63.3 | 69.5 | 74.2 |
| | SN (Koch et al., 2015) | - | - | - | - | - | 38.9 | - | 64.6 | - | 76.4 |

Table 1: Top-1 and top-5 accuracies (%) on the novel classes for the ImageNet based $n$-shot classification benchmark. We use ResNet-10 as the feature extractor. PN: prototypical networks, PMN: prototype matching networks, Cos-Cls: cosine classifiers. Methods with 'w/ G' use a meta-learned hallucinator. Standard deviations for all numbers are of the order of $0.2\%$. Our PECAN achieves the best performance. Importantly, PECAN is *model-agnostic* and can be combined with different meta-learning models to improve their performance.

| Method | $\ell^{cls}_{learner}$ | $\ell^{pre}_{learner}$ | $\ell^{cls}_{hal}$ | $\ell^{pre}_{hal}$ | Top-1 accuracy | | | | | Top-5 accuracy | | | | |
|---|---|---|---|---|---|---|---|---|---|---|---|---|---|---|
| | | | | | n=1 | 2 | 5 | 10 | 20 | n=1 | 2 | 5 | 10 | 20 |
| PN w/ G + PECAN | ✓ | | | | 19.9 | 26.4 | 35.7 | 41.9 | 45.9 | 45.0 | 55.9 | 67.3 | 73.0 | 76.5 |
| | | | ✓ | | 6.5 | 10.6 | 19.5 | 27.9 | 35.1 | 30.5 | 41.0 | 55.3 | 63.8 | 68.7 |
| | ✓ | | ✓ | | 20.1 | 26.8 | 36.7 | 42.6 | 47.0 | 45.3 | 56.0 | 67.5 | 73.3 | 76.8 |
| | ✓ | ✓ | | | 20.1 | 26.6 | 36.0 | 42.8 | 47.6 | 45.8 | 56.2 | 67.6 | 74.0 | 77.6 |
| | | | ✓ | ✓ | 6.8 | 10.8 | 19.7 | 27.9 | 35.2 | 31.5 | 41.4 | 55.4 | 63.8 | 68.7 |
| | ✓ | ✓ | ✓ | ✓ | **20.8** | **27.7** | **37.5** | **43.9** | **48.4** | **46.5** | **57.5** | **69.3** | **74.9** | **78.1** |

Table 2: Ablation on precision and collaboration requirements. '$\ell^{cls}$': hard precision based on classification loss, '$\ell^{pre}$': soft precision-inducing loss, '$\ell_{hal}$': collaborative objective imposed on the hallucinator. Different components are complementary to each other.

## 5.1 RESULTS ON IMAGENET

**Implementation details.** We mainly use a ResNet-10 architecture (He et al., 2016) as the feature extractor, following Hariharan & Girshick (2017); Wang et al. (2018). Additionally, we provide results using a deeper ResNet-50 architecture in Section A.3. We extract and record the features, and perform meta-learning by using these pre-computed features. We consider three widely-used, powerful meta-learning approaches: prototypical networks (PN) (Snell et al., 2017), prototype matching networks (PMN) (Vinyals et al., 2016; Wang et al., 2018), and cosine classifiers (Cos-Cls) used in Gidaris & Komodakis (2018); Chen et al. (2019a). More implementation details are included in Section A.1.

**Baselines.** First we compare with the state-of-the-art meta-learning with hallucination method (Wang et al., 2018), which is a special case of our approach learned with only the hard precision loss. While Wang et al. (2018) focused on 'PN w/ G' and 'PMN w/ G', here we consider an additional type of classier with hallucination, 'Cos-Cls w/ G', to show the generality of our work. In addition, we compare with a variety of baselines, including (1) these meta-learning approaches with standard data augmentation techniques (Chen et al., 2019a); (2) data hallucination approaches which are not meta-learned: logistic regression with analogies hallucination (Hariharan & Girshick, 2017) and Gaussian hallucinator (Wang et al., 2018); (3) other recent meta-learning approaches: matching networks (MN) (Vinyals et al., 2016), model-agnostic meta-learning (MAML) (Finn et al., 2017), and 'cosine classifier & attentive weight generators (Cos & Att)' (Gidaris & Komodakis, 2018); (4) classical few-shot learning approaches: Siamese networks (SN) (Koch et al., 2015); and (5) simple baselines which are not meta-learned: logistic regression (Hariharan & Girshick, 2017). For fair comparison, all these baselines and our approach use the same pre-trained ConvNet backbone.

**Comparisons with the state of the art.** Table 1 shows that our PECAN *consistently* outperforms all the baselines *by large margins across different scenarios*. For this challenging 311-way classification, our improvements are of the order of $1\%$ to $2\%$, while standard deviations for accuracy are of the order of $0.2\%$. For example, in the case of top-5 accuracy, our 'PN w/ G + PECAN' outperforms 'PN w/ G' by 1.5 points for $n = 1$ and 1.6 points for $n = 20$. Similar trends can be observed for 'PMN w/

G + PECAN' and 'Cos-Cls w/ G + PECAN', and also in the top-1 accuracy regime. This indicates that our approach is general and can work with different meta-learners.

**Ablation studies.** We conduct a series of ablations to evaluate the contribution of each component and different design choices. We use the prototypical network (PN) here due to its fast training speed.

| | | | Accuracy (%) | |
|---|---|---|---|---|
| Method | | | $n$=1 | 5 |
| Matching Networks (Vinyals et al., 2016) | | | 43.56± 0.84 | 55.31±0.73 |
| MAML (Conv-4) (Finn et al., 2017) | | | 48.70± 1.84 | 63.11±0.92 |
| Prototypical Networks (Snell et al., 2017) | | | 49.42± 0.78 | 68.20±0.66 |
| Reptile (Nichol & Schulman, 2018) | | | 49.97±0.32 | 65.99±0.58 |
| Meta-SGD (Li et al., 2017) | | | 50.47± 1.87 | 64.03±0.94 |
| MetaGAN (Zhang et al., 2018a) | | | 52.71±0.64 | 68.63±0.67 |
| Baseline++ (Chen et al., 2019a) | | | 53.97±0.79 | 76.16±0.63 |
| Cos & Att. (Gidaris & Komodakis, 2018) | | | 55.45± 0.89 | 70.13 ±0.68 |
| SNAIL (Mishra et al., 2018) | | | 55.71± 0.99 | 68.88±0.92 |
| Relation Networks (Sung et al., 2018) | | | 57.02± 0.92 | 71.07±0.69 |
| TADAM (Oreshkin et al., 2018) | | | 58.50±0.30 | 76.70±0.30 |
| Delta-Encoder (Schwartz et al., 2018) | | | 58.7 | 73.6 |
| IDeMe-Net (Chen et al., 2019b) | | | 59.14± 0.86 | 74.63±0.74 |
| LEO (Rusu et al., 2019) | | | 61.76±0.08 | 77.59±0.12 |
| MetaOptNet-SVM (Lee et al., 2019) | | | 62.64±0.61 | 78.63±0.46 |
| SalNet (Zhang et al., 2019) | | | 57.45±0.88 | 72.01±0.67 |
| EGNN+Transduction (Kim et al., 2019) | | | - | 76.37 |
| MAML (ResNet-10) (Finn et al., 2017) | | | 54.69±0.89 | 66.62±0.83 |
| MAML w/ G (Wang et al., 2018) | | | 56.37±0.63 | 68.91±0.57 |
| MAML w/ G + PECAN (Ours) | | | 58.39±0.37 | 71.36±0.44 |
| PMN w/ G (Wang et al., 2018) | | | 62.28±0.53 | 78.28±0.62 |
| PMN w/ G + PECAN (Ours) | | | **63.93±0.40** | **80.58±0.29** |

| | | Top-5 accuracy | | | |
|---|---|---|---|---|---|
| Similarity measure | $n$=1 | 2 | 5 | 10 | 20 |
| $-\cos(p^{\text{real}}, p^G)$ | 44.9 | 55.8 | 66.8 | 72.5 | 76.0 |
| $\text{CE}(p^{\text{real}}, p^G)$ | 44.2 | 53.9 | 64.5 | 70.4 | 73.7 |
| $\text{CE}(\widehat{p}^{\text{real}}, \widehat{p}^G)$ | 44.0 | 53.9 | 65.0 | 70.7 | 74.1 |
| $\text{JS}(p^{\text{real}}, p^G)$ | 44.9 | 55.6 | 66.9 | 72.8 | 76.2 |
| $\text{JS}(\widehat{p}^{\text{real}}, \widehat{p}^G)$ | 45.0 | 56.0 | 67.4 | 73.3 | 76.6 |
| $\text{sKL}(p^{\text{real}}, p^G)$ | 44.9 | 55.6 | 67.0 | 72.7 | 76.2 |
| $\text{sKL}(\widehat{p}^{\text{real}}, \widehat{p}^G)$ | 45.0 | 55.9 | 67.1 | 72.8 | 76.2 |
| $-\cos(\widehat{p}^{\text{real}}, \widehat{p}^G)$ | **45.8** | **56.2** | **67.6** | **74.0** | **77.6** |

Table 3: Ablation on choice of similarity measure in the soft precision-inducing loss. $p^{\text{real}}$ and $p^G$: class probabilities of $h^{\text{real}}$ and $h^G$, respectively. $\widehat{p}^{\text{real}}$ and $\widehat{p}^G$: class probabilities in the absence of the ground-truth labels. 'CE': cross-entropy loss as in knowledge distillation (Hinton et al., 2015), 'JS': Jensen-Shannon divergenc, 'sKL': symmetric KL-divergence. Our similarity measure achieves the best performance.

Table 4: Test accuracies (%) on the novel classes for the *mini*ImageNet dataset. '±' indicates 95% confidence intervals over tasks. Our PECAN significantly outperforms the state-of-the-art approaches.

*Variants of PECAN.* PECAN leverages two requirements for the meta-learned hallucinator: precision and collaboration. '$\ell^{\text{cls}}_{\text{learner}}$' is the basic hallucinator with only the hard precision based on the classification loss. Table 2 shows that each requirement by itself yields performance superior to the basic hallucinator. The soft precision-inducing loss $\ell^{\text{pre}}$ consistently helps when combined with the hard precision $\ell^{\text{cls}}$: '$\ell^{\text{cls}}_{\text{learner}} + \ell^{\text{pre}}_{\text{learner}}$' outperforms '$\ell^{\text{cls}}_{\text{learner}}$' and '$\ell^{\text{cls}}_{\text{hal}} + \ell^{\text{pre}}_{\text{hal}}$' outperforms '$\ell^{\text{cls}}_{\text{hal}}$'. The collaboration objective integrates $\ell_{\text{learner}}$ and $\ell_{\text{hal}}$ to boost the performance: '$\ell^{\text{cls}}_{\text{learner}} + \ell^{\text{cls}}_{\text{hal}}$' outperforms '$\ell^{\text{cls}}_{\text{learner}}$'. Each component is thus essential and complementary to each other, enabling our full PECAN to outperform its variants.

*Choice of similarity measure in soft precision-inducing loss.* Our precision-inducing loss measures the similarity between classifier predictions $p^{\text{real}}$ and $p^G$. We used negative cosine distance between the probabilities $\widehat{p}^{\text{real}}$ and $\widehat{p}^G$ in the absence of ground-truth labels. Table 3 compares with other types of similarity: variant of negative cosine distance, cross-entropy as in knowledge distillation (Hinton et al., 2015), Jensen-Shannon divergence, and symmetric KL-divergence (Dvornik et al., 2019). Our similarity achieves the best performance, and removing the true-class probability consistently helps.

*Impact of collaborative objective.* Our collaborative objective introduces additional direct and early supervision to train the hallucinator. Table 2 shows quantitatively its contribution to the overall accuracy. Here, we further qualitatively understand its impact though t-SNE visualizations (van der Maaten & Hinton, 2008) of the hallucinated examples for novel classes. For ease of analysis, we do not use the precision-inducing loss. Without the collaborative objective, despite being able to match the class distributions in the embedding space $\Phi$ (Figure 3b), the hallucinated examples are initially pulled away from the class distributions in the pre-trained feature space $\mathcal{X}$ (Figure 3a), indicating a "lazy" hallucinator. In contrast, the collaborative objective enforces the hallucinator to generate more discriminative examples *right after hallucination* (Figure 3c), leading to improved performance.

**Qualitative visualizations.** To better understand the hallucination process, Figure 4 shows some examples of classification results for our PECAN and the state-of-the-art baseline (Wang et al., 2018).

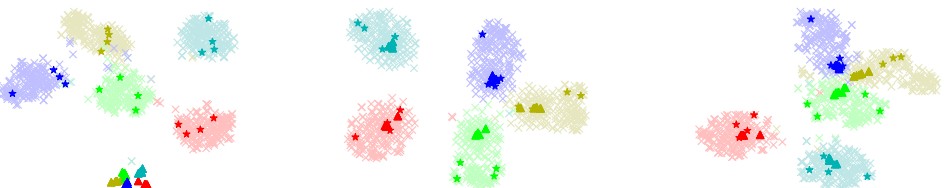

(a) In pre-trained space $\mathcal{X}$ w/o CO    (b) In embedding space $\Phi$ w/o CO    (c) In pre-trained space $\mathcal{X}$ w/ CO

Figure 3: t-SNE visualizations of hallucinated examples for investigating the impact of collaborative objective (CO) without precision. Seeds are shown as stars, real examples as crosses, hallucinations as triangles. PECAN *without* CO: (a) in the pre-trained ResNet-10 feature space $\mathcal{X}$, (b) in the new embedding space $\Phi$ learned by PN; PECAN *with* CO: (c) in the pre-trained ResNet-10 feature space $\mathcal{X}$. **Best viewed in color with zoom.**

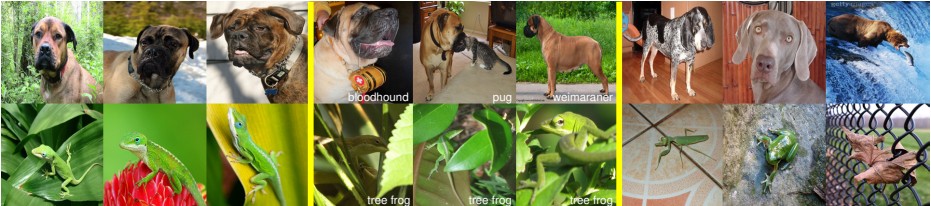

Figure 4: Visual comparisons of top-1 classification results on two representative novel classes between our PECAN and the state-of-the-art meta-learned hallucinator (Wang et al., 2018). Top row: bullmastiff; bottom row: American chameleon. Left 3 columns: test images that are correctly classified by both approaches; middle 3 columns: target test images that are misclassified by Wang et al. (2018) as other classes (the names of the predicted classes by Wang et al. (2018) are overlaid on the images), but correctly classified by PECAN; right 3 columns: test images from other classes that are misclassified by Wang et al. (2018) as the target class, but correctly classified by PECAN. Our approach is able to model a large range of visual variations and diversity, e.g., bullmastiffs in different poses, and chameleons in different viewpoints and background, whereas Wang et al. (2018) is confused by visually similar classes.

## 5.2 RESULTS ON *mini*IMAGENET

To show the generality of our approach, we further evaluate on *mini*ImageNet. We use a ResNet-10 architecture and focus on incorporating our hallucinator into the metric-learning-based meta-learning approach, prototype matching networks (PMN) (Wang et al., 2018), and the optimization-based meta-learning approach, MAML (Finn et al., 2017). For MAMl, in each meta-training episode, we sample a batch of few-shot classification tasks. For each of the tasks, we sample training sets $S_{\text{train}}^*$ and $S_{\text{train}}$, sample a test test $S_{\text{test}}$, hallucinate $S_{\text{train}}^G$, and obtain an augmented training set $S_{\text{train}}^{\text{aug}}$. In the MAML inner loop, for each task, conditioning on its $S_{\text{train}}^{\text{aug}}$ ($S_{\text{train}}^G$, or $S_{\text{train}}^*$), we adapt the parameters of $h^{\text{cls}}$ ($h^G$, or $h^{\text{real}}$) using few gradient updates. For each task, the adapted $h^{\text{cls}}$ ($h^G$, or $h^{\text{real}}$) is evaluated on the corresponding $S_{\text{test}}$. In the MAML outer loop, we average the classification objective on $S_{\text{test}}$ across the batch of tasks. In a similar way, we compute the collaborative objective in the pre-trained feature space. The final loss is used to update the initial MAML model and the hallucinator. From Table 4, our PECAN significantly outperforms all these state-of-the-art competitors, including other hallucination based approaches such as MetaGAN (Zhang et al., 2018a), delta-encoder (Schwartz et al., 2018), IDeMe-Net (Chen et al., 2019b), and SalNet (Zhang et al., 2019). Our superior performance over MetaGAN, a GAN-based approach to hallucinate data, shows that directly matching the classification performance is more desirable than matching the data distribution between hallucinated and real examples for recognition tasks. Our generic framework can be combined with more recent meta-learning methods, such as LEO (Rusu et al., 2019) and MetaOptNet-SVM (Lee et al., 2019), for further improvement.

## 6 CONCLUSION

We have presented an approach to few-shot classification that uses a precise collaborative hallucinator to generate additional examples. Our hallucinator integrates two important requirements that facilitate data hallucination in a way that most improves the classification performance, and is trained end-to-end through meta-learning. The extensive experiments demonstrate our state-of-the-art performance on the challenging ImageNet and *mini*ImageNet based few-shot benchmark in various scenarios.

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

# A APPENDIX

## A.1 ADDITIONAL IMPLEMENTATION DETAILS

The embedding architecture of PN is composed of two-layer MLPs with leaky ReLU nonlinearity of slope $0.01$, and a Euclidean distance similar to Snell et al. (2017). The embedding architecture of PMN is composed of a one-layer bi-directional LSTM and attention LSTM, and a cosine distance as in (Vinyals et al., 2016; Wang et al., 2018). The embedding architecture of Cos-Cls is composed of two-layer MLPs with leaky ReLU nonlinearity of slope 0.01 and an additional one-layer MLP without nonlinearity, and a cosine distance with a learnable temperature similar to Gidaris & Komodakis (2018). The initial value of the temperature is 100. The classification weight vector is estimated by averaging the feature vectors of the training examples for each class. The hallucinator $G$ is a three-layer MLP with leaky ReLU nonlinearity of slope $0.01$, with its parameters initialized to block diagonal identity matrices (Wang et al., 2018). The dimensionality of the hidden layers is $512$ for ResNet-10 features and $2,048$ for ResNet-50 features.

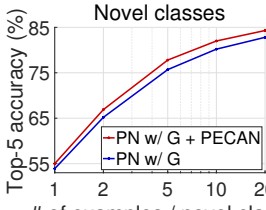

| Precision inducing | | Top-5 accuracy | | | | |
| --- | --- | --- | --- | --- | --- | --- |
| | $n$=1 | 2 | 5 | 10 | 20 |
| $\|\|m^{\text{real}} - m^{G})\|\|_2$ | | 45.3 | 56.0 | 67.6 | 73.2 | 76.6 |
| $-\cos(\widehat{p}^{\text{real}}, \widehat{p}^{G})$ (**Ours**) | | **45.8** | **56.2** | **67.6** | **74.0** | **77.6** |

Figure A.1: Top-5 accuracy (%) on the novel classes for the ImageNet based $n$-shot classification benchmark. We use ResNet-50 as the feature extractor. PN: prototypical networks. Methods with 'w/ G' use a meta-learned hallucinator. With a deeper network, all accuracies are higher, and our PECAN significantly outperforms the baselines.

Table A.1: Additional analysis of the soft precision-inducing loss in the case of PN on the ImageNet based $n$-shot classification benchmark. $\widehat{p}^{\text{real}}$ and $\widehat{p}^{G}$: class probabilities of $h^{\text{real}}$ and $h^{G}$ in the absence of the ground-truth labels, respectively. $m^{\text{real}}$ and $m^{G}$: class means of real and hallucinated examples, respectively. Our *generic* similarity generalizes significantly better than the specific similarity applicable to PN for novel calsses.

Both the baselines (PN/PMN/Cos-Cls and PN/PMN/Cos-Cls with hallucination) and our approach are meta-trained on ImageNet for 60,000 episodes by SGD with an initial learning rate 0.05 for PN/PMN and 0.005 for Cos-Cls, decayed by a factor of 10 every 20,000 episodes. In each episode of the meta-training stage, we sample all the base classes following Wang et al. (2018), which found that it is advantageous to use more classes rather than fewer. We sample $n^*$=20 examples per class from the base dataset $D_{\text{base}}$, leading to an initial training set $S^*_{\text{train}}$. Consistent with Wang et al. (2018), to make a single hallucinator robust to different sample sizes, we randomly sample different sized examples per class from $S^*_{\text{train}}$, from 1 to 15 examples per class, to obtain $S_{\text{train}}$. One random seed example is sampled from $S_{\text{train}}$, and fed into the hallucinator $G$ with different noise vectors to generate 20 examples per class as $S^G_{\text{train}}$. Hallucinated examples are sampled from $S^G_{\text{train}}$ and added to $S_{\text{train}}$ until there are exactly 20 examples per class in $S^{\text{aug}}_{\text{train}}$. For the test set $S_{\text{test}}$, we have 5 random examples per class for prototypical networks (PN) (Snell et al., 2017) and cosine classifiers(Cos-Cls) (Gidaris & Komodakis, 2018), and 1 random example per class for prototype matching networks (PMN) (Vinyals et al., 2016; Wang et al., 2018).

For the soft precision-inducing loss, we use softmax with temperature to produce conditional class probabilities $\widehat{p}^{\text{real}}$ and $\widehat{p}^{G}$. This temperature is a learnable parameter. It is shared between $\widehat{p}^{\text{real}}$ and $\widehat{p}^{G}$, but is not shared between $\mathcal{L}_{\text{learner}}$ and $\mathcal{L}_{\text{hal}}$.

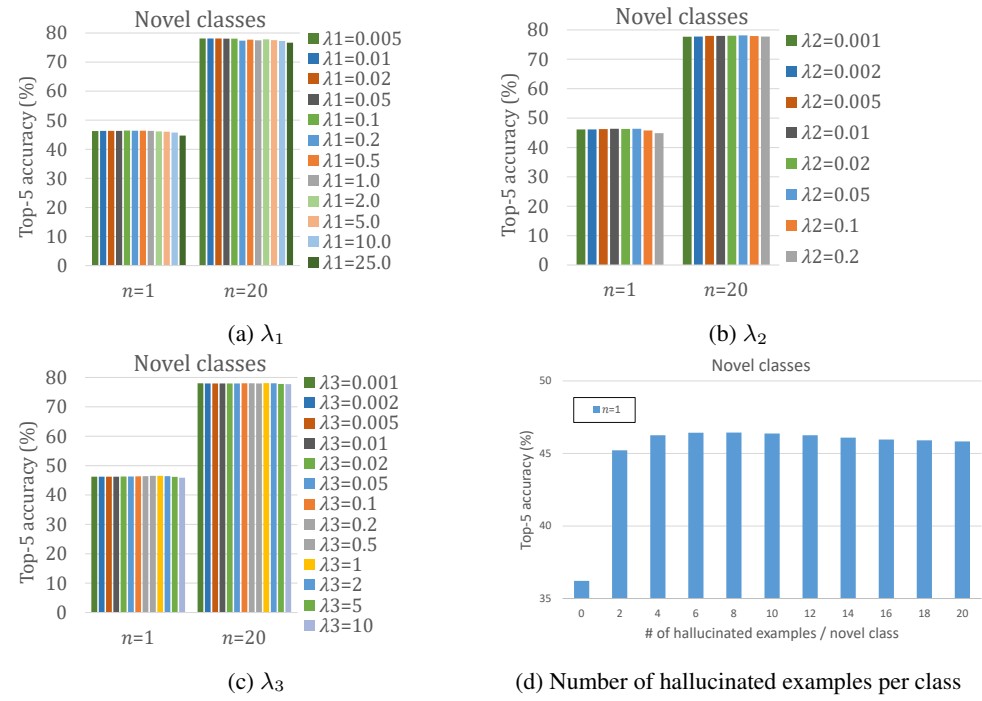

Figure A.2: Top-5 accuracy (%) of additional hyper-parameter analysis on the novel classes for the ImageNet based $n$-shot classification benchmark. This analysis is conducted on 'PN w/ G + PECAN' with ResNet-10. (a)(b)(c) provide sensitivity analysis of the hyper-parameters $\lambda_1$, $\lambda_2$, and $\lambda_3$ in the overall objective. The performance of our PECAN is stable over a wide range of hyper-parameter values. (d) shows how our performance gradually increases and then saturates as more examples are hallucinated for $n = 1$-shot classification.

The hyper-parameters obtained by cross-validation on ImageNet are: $\lambda_1$=0.5, $\lambda_2$=0.02, and $\lambda_3$=0.1 for PN with ResNet-10; $\lambda_1$=0.2, $\lambda_2$=0.0001, and $\lambda_3$=0.05 for PN with ResNet-50; $\lambda_1$=0.4, $\lambda_2$=2.0, and $\lambda_3$=0.1 for PMN with ResNet-10; and $\lambda_1$=0.5, $\lambda_2$=0.000001, and $\lambda_3$=0.005 for the cosine classifier with ResNet-10.

During the meta-testing stage, following Wang et al. (2018) we use $n$=1, 2, 5, 10 or 20 examples per class from the novel dataset $D_{\mathrm{novel}}$, and then hallucinate a fixed number of additional examples for each novel class. By cross-validation, the number of hallucinated examples per class is set to 10 for PN, Cos-Cls and Cos-Cls w/ G + PECAN with ResNet-10, 8 for Cos-Cls w/ G with ResNet-10, 5 for PN with ResNet-50, and 20 for PMN with ResNet-10. We combine the classifier prediction results in the pre-trained feature space and the learned embedding space using a scalar hyper-parameter. By cross-validation, this hyper-parameter is set to 0.05 for PN with ResNet-10, 0.07 for PN with ResNet-50, 1 for PMN with ResNet-10, and 0.00002 for the cosine classifier with ResNet-10. For the test set $S_{\mathrm{test}}$, following Wang et al. (2018) we have 50 real examples per class, and we average the top-1 or top-5 accuracy of them over the novel classes.

## A.2 PERFORMANCE ON BASE CLASSES

While we significantly improve the classification performance on novel classes, our approach remains accurate on base classes. For example, for prototypical networks (PN) (Snell et al., 2017), both our 'PN w/ G + PECAN' and the baseline 'PN w/ G' achieve the same top-5 accuracy 92.4% on ImageNet.

## A.3 IMPACT OF DEEPER REPRESENTATION MODELS

Figure A.1 shows the results on ImageNet using features from a ResNet-50 architecture. As expected, deeper networks result in better performance for all the approaches, but our PECAN hallucination

strategy still provides large gains across the board over the state-of-the-art meta-learned hallucinator in (Wang et al., 2018).

## A.4    ANALYSIS OF HYPER-PARAMETER SENSITIVITY

**Hyper-parameters in the overall objective.**    We conduct sensitivity experiments for the hyper-parameters $\lambda_1$, $\lambda_2$, and $\lambda_3$, which trade off different loss components in the overall objective of our PECAN. We vary one of the three hyper-parameters while fixing the remaining two to their cross-validated values. Figures A.2a, A.2b, and A.2c show the top-5 accuracy of 'PN w/ G + PECAN' on the novel classes for the ImageNet based $n$-shot classification benchmark. We can see that the top-5 accuracy is stable *over a wide range* of hyper-parameter values, for example when the value of $\lambda_1$ becomes 50 times larger or 100 times smaller than $\lambda_1$ used in the main paper. Across the board, our PECAN consistently and significantly outperforms the baselines shown in the main paper.

**Number of hallucinated examples.**    We also show how the top-5 accuracy changes for $n = 1$-shot classification with respect to the number of hallucinated images in Figure A.2d. We can see that when the number of hallucinated examples is changed from 0 to 10, the performance of our PECAN gradually improves, and then saturates and drops slightly with more than 10 images generated.

## A.5    ADDITIONAL ANALYSIS OF SOFT PRECISION-INDUCING LOSS

Our soft precision-inducing loss measures the similarity between *classifier predictions* $p^{\mathrm{real}}$ and $p^G$. This is a *general* similarity measure which applies to various types of classifiers, including parametric and non-parametric classifiers. For prototypical networks (PN) (Snell et al., 2017), a non-parametric nearest centroid classifier is used to assign class probabilities for a test example based on its distances from class means. Hence, in this special case, to measure the similarity between classifier predictions, we can directly calculate the distance between the mean of real examples $m^{\mathrm{real}}$ and the mean of hallucinated examples $m^G$. Table A.1 compares our generic similarity with this specific similarity for PN on the ImageNet based $n$-shot classification benchmark. The result shows that our similarity generalizes significantly better for novel classes.

## A.6    ADDITIONAL VISUALIZATIONS OF CLASSIFICATION RESULTS

Similar to Figure 5 in the main paper, here we provide more examples of classification results for our PECAN and the state-of-the-art meta-learned hallucinator (Wang et al., 2018) in Figure A.3.

## A.7    EXTENSION TO FEW-SHOT REGRESSION

In the main paper, we focus on few-shot classification tasks. However, our approach is general and applies to few-shot regression tasks as well. When extending our approach to address regression tasks, we need to slightly modify the design of the hallucinator $G$. Specifically, in classification tasks, the hallucinator $G$ takes as input a seed example $(x, y)$ and outputs a hallucinated example $(x', y') = (G(x, z), y)$, where the class label $y' = y$, indicating that the hallucinated example belongs to the same class as the seed example. In contrast, in regression tasks, $y$ becomes a continuous quantity. We thus cannot enforce the constraint $y' = y$. To address this issue, we modify the hallucinator G to directly generate the tuple $(x', y') = G(x, y, z)$. This is achieved by the concatenation of $x$ and $y$ as input to $G$.

We incorporate our hallucinator with MAML (Finn et al., 2017) and evaluate on the $n = 5$-shot sinusoidal regression task proposed in Finn et al. (2017). Each task regresses from the input to the output of a sine curve, and different tasks differ in the amplitude and phase of the sinusoid. The amplitude and the phase are uniformly distributed within $[0.1, 5.0]$ and $[0, \pi]$, respectively. During meta-training and meta-testing, the input datapoints $x$ of each task are uniformly sampled from $[-5.0, 5.0]$. The prediction loss is measured by the mean squared error (MSE) between the prediction and true value. The regressor is a feedforward neural network with 2 hidden layers of size 40 with ReLU nonlinearity. We use MAML (Finn et al., 2017) with one gradient update based on $n = 5$ examples with a fixed step size $\alpha = 0.01$, and use Adam as the meta-optimizer (Kingma & Welling, 2014).

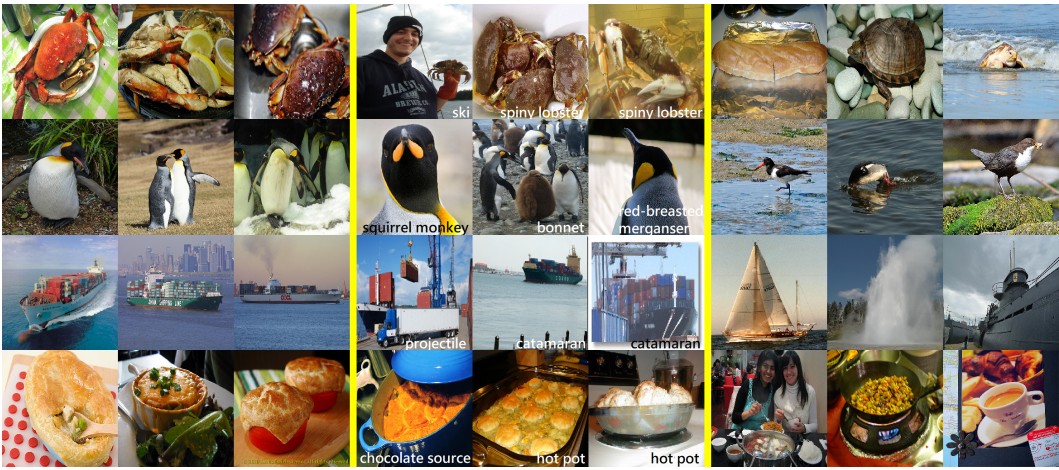

Figure A.3: Additional visual comparisons of top-1 classification results on four representative novel classes between our PECAN and the state-of-the-art meta-learned hallucinator (Wang et al., 2018). From top to bottom rows: Dungeness crab, king penguin, container ship, and potpie. Left 3 columns: test images that are correctly classified by both approaches; middle 3 columns: target test images that are misclassified by Wang et al. (2018) as other classes (the names of the predicted classes by Wang et al. (2018) are overlaid on the images), but correctly classified by PECAN; right 3 columns: test images from other classes that are misclassified by Wang et al. (2018) as the target class, but correctly classified by PECAN. Our approach is able to model a large range of visual variations and diversity, whereas Wang et al. (2018) is confused by visually similar classes.

| Method | mean squared error (MSE) n=5 |
| --- | --- |
| MAML (Finn et al., 2017) | 0.84 |
| MAML w/ G (Wang et al., 2018) | 0.67 |
| MAML w/ G + PECAN (Ours) | **0.52** |

Table A.2: Comparison on the $n = 5$-shot sinusoidal regression task in Finn et al. (2017). Our PECAN is general and outperforms the baselines for the regression task as well.

We compare with two baselines: (1) standard MAML, and (2) 'MAML w/ G', which is MAML with the hallucinator in (Wang et al., 2018). While Wang et al. (2018) focus on classification tasks, here we extend the use of its hallucinator in the similar way as we discussed before. Table A.2 shows that our PECAN consistently outperforms the baselines for the regression task as well, indicating the generality of our approach.

## A.8 VISUALIZATIONS OF HALLUCINATED EXAMPLES

Our hallucination is performed in the pre-trained feature space, and visualizing them directly is not intuitive. In addition to t-SNE visualization of hallucinated samples in Figure 3, here in Figure A.4 we include an additional visualization of hallucinated examples in the pixel space, using the nearest neighbor real image in the feature space, corresponding to each hallucinated feature sample.

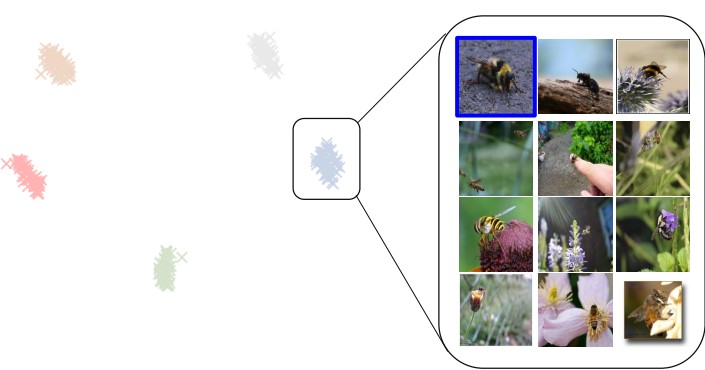

Figure A.4: Hallucinated example visualization for 5 novel classes. Left: t-SNE visualization of hallucinated examples in the feature space. Right: The single image framed in blue is the sampled seed example. All other images represent the hallucinated examples which are visualized by using their nearest neighbor real image in the feature space.

