# OpenReview forum: "Meta-Learning by Hallucinating Useful Examples"
_ICLR.cc/2020/Conference — Reject_

### Official Review · AnonReviewer2 · 2019-10-09
**Official Blind Review #2**

**Rating:** 3

**Review:**

In this paper, the authors address few-shot learning via a precise collaborative hallucinator. In particular, they follow the framework of (Wang et al., 2018), and introduce two kinds of training regularization. The soft precision-inducing loss follows the spirit of adversarial learning, by using knowledge distillation. Additionally, a collaborative objective is introduced as middle supervision to enhance the learning capacity of hallucinator.

Here are some comments:
1. The novelty is relatively limited. The idea of hallucinating has been mainly introduced in (Wang et al., 2018). The soft precision-inducing loss is a straightforward extension of knowledge distillation (Hinton et al., 2015).

2. It is not quite clear about how to use the collaborative objective on the hallucinator. Fig.2 and the text in ' Collaboration between hallucinator and learner ' (page 5) are not quite informative. Especially, how to perform the soft precision-inducing loss (l_hal^pre) for hallucinator?

**Experience Assessment:**

I have published one or two papers in this area.

**Review Assessment: Checking Correctness Of Derivations And Theory:**

I assessed the sensibility of the derivations and theory.

**Review Assessment: Checking Correctness Of Experiments:**

I assessed the sensibility of the experiments.

**Review Assessment: Thoroughness In Paper Reading:**

I read the paper at least twice and used my best judgement in assessing the paper.

---

> ### Author Response · Authors · 2019-11-13
> **Response to Review #2**
>
> We thank the reviewer for the comments. The comments focus mostly on the novelty and the clarity of some details. We address all these points as follows.
>
> 1. The novelty ... hallucinating in Wang et al., 2018 ... knowledge distillation (Hinton et al., 2015):
>
> While we agree with the reviewer that our approach falls into the category of meta-learning with data hallucination (Wang et al., 2018), it is not a trivial extension to any of the existing work. Different from Wang et al., 2018, which learns the data hallucinator solely through end-to-end training with the classifier and without any additional constraints, we investigated critical and unexplored properties (i.e., precision and collaboration) that the data hallucinator should satisfy. We instantiated these properties as novel loss functions to improve the precision of the hallucinated examples, and enhance the collaboration between the hallucinator and the classifier. In addition, from a broader perspective, our approach not only extracts shared knowledge across a collection of few-shot learning tasks as most of the existing meta-learning approaches normally do, but also leverages additional knowledge in large-sample models to guide hallucination and few-shot learning. As we have shown in the experiments, our approach consistently and significantly outperforms Wang et al., 2018 for different meta-learning approaches and across different datasets.
>
> While our soft precision-inducing loss is related to knowledge distillation (Hinton et al., 2015), as mentioned in Section 4, paragraph “Soft precision-inducing hallucinator”, page 5, it is different in three important ways. First, knowledge distillation typically focuses on model compression, which trains a shallow “student” neural network by mimicking the output of a deep “teacher” model. These two models are trained on the same data but are of different capacities. In contrast, our soft precision-inducing loss compares two models of the same capacity but trained on different types of data: a classifier trained on real data, and another trained on hallucinated examples. Second, our formulation measures the similarity of two probabilities in the absence of ground-truth labels, which is different from the formulation in Hinton et al., 2015. Third, in the ablation study of “Choice of similarity measure in soft precision-inducing loss” and Table 3 (page 8), we empirically showed that our loss function significantly outperformed the loss used in Hinton et al., 2015, as well as other similarity measures.
>
> 2. how to use the collaborative objective on the hallucinator … the soft precision-inducing loss (l_hal^pre) for hallucinator:
>
> The soft precision-inducing losses are computed in a similar way for l_hal^pre and l_learner^pre. The difference lies in that l_learner^pre is a loss in the classifier embedding space, while l_hal^pre is in the pre-trained feature space. This is also noted in Review #1: the collaborative objective applies “direct early supervision in the feature space in which the hallucination is conducted in addition to in the classifier embedding space.”
>
> More precisely, we first pre-train a deep convolutional network using a standard cross-entropy loss, and use it to extract image features. Meta-learning is then performed over these pre-trained features (the last paragraph in Section 3, page 4). Without loss of generality, we take prototypical networks (PN) as an example (the last paragraph in Section 4, page 6). PN learns a new embedding space Φ on top of the pre-trained feature space X, and uses a non-parametric nearest centroid classifier to assign class probabilities for a test example based on its distances from class means in Φ. The embedding architecture of PN is composed of two-layer MLPs (implementation details in Section A.1).
>
> In each meta-training episode, after sampling S_train^∗, S_train, and S_test and hallucinating S_train^G in the pre-trained feature space X, we perform nearest centroid classification, and produce the collaborative objective L_hal on S_test, including the soft precision-inducing loss l_hal^pre. We then feed the examples to the PN learner, obtain their embedded features in Φ, perform nearest centroid classification, and produce the classification objective L_learner on S_test, including the soft precision-inducing loss l_learner^pre (the last paragraph in Section 4, page 6).

---

### Official Review · AnonReviewer3 · 2019-10-22
**Official Blind Review #3**

**Rating:** 6

**Review:**

This paper proposes a general meta-learning with hallucination framework called PECAN. It is model-agnostic and can be combined with any meta-learning models to consistent boost their few-shot learning performance.

There are two key points for the proposed model. On the one hand, the authors introduce a novel precision-inducing loss which encourages the hallucinator to generate examples so that a classifier trained on them makes predictions similar to the one trained on a large amount of real examples. On the other hand, the authors introduce a collaborative objective for the hallucinator as early supervision, which directly facilitates the generation process and improves the cooperation between the hallucinator and the learner.

On the whole, the paper is well-written, and the proposed idea is novel and interesting.

I have some following major concerns about the paper:
(1) In Figure 2, the authors first sample the training set S^*_{train}, which contains n^* examples for each of the m classes, and then they randomly sample n examples per class, and obtain a subset S_{train}. Why not generate the S_{train} directly and then measure your precision-inducing loss over the real set S_{train} and S^G_{train}? I hope the authors explain it in their paper.
(2) For Function 2 in the paper, why compute the cosine distance on the probability vectors that are obtained by removing the logit for ground-truth label in original probability distributions? Could we compute the distance on the probability vectors that contains the logit for ground-truth label? I hope the authors explain it in detail.
(3) As far as I know, there are some latest work on few-shot learning in 2019, especially the work “Few-shot Learning via Saliency-guided Hallucination of Samples” and “Edge-Labeling Graph Neural Network for Few-shot Learning”. I hope the authors can compare with these two methods to further demonstrate the effectiveness of the proposed model.

**Experience Assessment:**

I have published one or two papers in this area.

**Review Assessment: Checking Correctness Of Derivations And Theory:**

I carefully checked the derivations and theory.

**Review Assessment: Checking Correctness Of Experiments:**

I carefully checked the experiments.

**Review Assessment: Thoroughness In Paper Reading:**

I read the paper thoroughly.

---

> ### Author Response · Authors · 2019-11-14
> **Response to Review #3**
>
> We thank the reviewer for the comments. The comments focus mostly on the clarity of some details and additional comparison. We address all these points as follows.
>
> (1) In Figure 2 … why not generate the S_{train} directly and then measure your precision-inducing loss over the real set S_{train} and S^G_{train}:
>
> This is because S^G_{train} is hallucinated from S_{train}. If we measure the precision-inducing loss over S^G_{train} and S_{train}, we will then end up with a trivial hallucinator which only needs to memorize S_{train} and is not useful for improving classification performance. In contrast, in our approach, we measure the precision-inducing loss over S^G_{train} and a much larger real set S^*_{train}. By doing so, the hallucinator is forced to generate additional useful examples from the small training set S_{train}, so that the classifier trained on them matches the performance of a classifier trained on a larger set of real examples S^*_{train}. Please refer to Section 1, paragraph 4, page 2, and Section 4, paragraph “Soft precision-inducing hallucinator”, page 5, for more details.
>
> In fact, introducing a large set of real examples S^*_{train} as learning guidance is one of our key contributions: we not only extract shared knowledge across a collection of few-shot learning tasks, but also leverage additional knowledge in large-sample models to guide hallucination and few-shot learning.
>
> (2) For Function 2 … could we compute the distance on the probability vectors that contains the logit for ground-truth label:
>
> Our final classification objective consists of the soft precision-inducing loss (i.e., Function 2), which is measured in the absence of ground-truth labels, and the hard precision loss, which is the standard cross-entropy classification loss measured with respect to ground-truth labels only (the last two sentences below Function 2, in the paragraph “Soft precision-inducing hallucinator”, Section 4, page 5).
>
> By doing so, we: (i) separate the impacts of the predictions associated with ground-truth labels and other entries in prediction probability vectors, and (ii) ensure that the soft precision-inducing loss is not dominated by the predictions associated with ground-truth labels, given that they have already contributed to the hard precision loss.
>
> Empirically, in the ablation study of “Choice of similarity measure in soft precision-inducing loss” and Table 3 (page 8), we showed that our loss function without ground-truth labels consistently and significantly outperformed the loss with ground-truth labels, for the cosine as well as other measures.
>
> (3) Some latest work … “Few-shot Learning via Saliency-guided Hallucination of Samples” (Ref1) and “Edge-Labeling Graph Neural Network for Few-shot Learning” (Ref2) … compare with these two methods:
>
> We have included the comparison with these two methods on miniImageNet in Table 4, and updated the submission accordingly. Our approach significantly outperforms these methods. Here we summarize the results:
>
> Method                                             n=1                 n=5
> SalNet (Ref1)                              57.45±0.88     72.01±0.67
> EGNN+Transduction (Ref2)             NA                 76.37
> Ours                                            63.93±0.40     80.58±0.29

---

### Official Review · AnonReviewer1 · 2019-10-27
**Official Blind Review #1**

**Rating:** 6

**Review:**

This paper describes a method that builds upon the work of Wang et al. It meta-learns to hallucinate additional samples for few-shot learning for classification tasks. Their two main insights of this paper are to propose a soft-precision term which compares the classifiers' predictions for all classes other than the ground truth class for both a few-shot training set and the hallucinated set and b) to introduce the idea of applying direct early supervision in the feature space in which the hallucination is conducted in addition to in the classifier embedding space. This allows for stronger supervision and prevents the hallucinated samples from not being representative of the classes. The authors show small, but consistent improvement in performance on two benchmarks: ImageNet and miniImageNet with two different network architectures versus various state-of-the-art meta-learning algorithms with and without hallucination. The authors have adequately cited and reviewed the existing literature. They have also conducted many experiments (both in the main paper and in the supplementary material) to show the superior performance of their approach versus the existing ones. Furthermore their ablation studies both for the type of soft precision loss and for their various individual losses are quite nice and thorough.

Overall the contribution of this paper is incremental over Wang et al and is mainly in the introduction of their new loss terms to regularize the hallucination process. This is clearly evident from Table 2 (comparing rows 1 and 3), where much of the performance gain is attained by including the l_learner^cls term versus the collaborative loss term (comparing row 1 and row 4).

Furthermore, I would like the author to answer the following two questions:

1. The authors claim that their method is general applicable to all meta-learning methods and can be combined with them. Yet, the meta learning methods that they apply it to: prototypical networks, prototypical matching networks and cosine classifiers are all metric-learning-based meta-learning techniques. I would like the authors to outline the procedure (and preferably also show experiments) for applying their proposed technique to meta-learning based techniques that do not involve learning a metric-embedding space and instead learn the learning procedure via nested optimization, e.g. MAML and its variants.

2. In Table 4, I would like to see the results of PNM w/G or in other words the results of (Wang et al, 2018)'s method in comparison to the authors' proposed method.

3. The authors make no attempt to solve the problem of hallucinating examples for regression tasks. This is fine as it is perhaps outside he scope of their current work. However, I would like the authors to fully qualify their claims everywhere in the paper and restrict the contribution of their work to classification tasks only.



**Experience Assessment:**

I have published in this field for several years.

**Review Assessment: Checking Correctness Of Derivations And Theory:**

N/A

**Review Assessment: Checking Correctness Of Experiments:**

I carefully checked the experiments.

**Review Assessment: Thoroughness In Paper Reading:**

I read the paper thoroughly.

---

> ### Author Response · Authors · 2019-11-15
> **Response to Review #1**
>
> We thank the reviewer for the comments. The comments focus mostly on the generality of our approach and additional comparison. We address all these points as follows.
>
> 1. … applying their proposed technique to meta-learning based techniques that do not involve learning a metric-embedding space and instead learn the learning procedure via nested optimization, e.g. MAML and its variant:
>
> In principle, our meta-learning with hallucination framework requires that the learner (i.e., the classification model h) is differentiable with respect to the examples in the augmented training set, thus allowing us to back-propagate the final loss and update not just the parameters of the classification model, but also the parameters of the hallucinator. This is true for many meta-learning algorithms, including the optimization-based meta-learning approaches such as MAML.
>
> To show the generality of our framework, we apply it to MAML. Implementation details of this integration are provided below. We have included the result of MAML, ‘MAML w/ G' which corresponds to hallucination-based MAML (i.e., hallucination framework following Wang et al., 2018), and ‘MAML w/ G + PECAN’ (i.e., our proposed hallucination framework) in Table 4, and updated the submission accordingly. As shown in the summary below, our approach is generic and applies to MAML as well.
>
> Method                                                               n=1                  n=5
> MAML (ResNet-10) (Finn et al., 2017)     54.69±0.89     66.62±0.83
> MAML w/ G (Wang et al., 2018)               56.37±0.63     68.91±0.57
> MAML w/ G + PECAN (Ours)                    58.39±0.37     71.36±0.44
>
> Implementation details: Following the procedure described in our submission, we use a ResNet-10 architecture as the feature extractor. We perform meta-learning over these pre-trained features. The embedding architecture of the MAML classification network h is composed of two-layer MLPs. The architecture of the generator G remains the same as in our submission.
>
> In each meta-training episode, we sample a batch of few-shot classification tasks. For each of the tasks, following the process in Figure 2, we sample training sets S_train^∗ and S_train, sample a test test S_test, hallucinate S_train^G, and obtain an augmented training set S_train^aug. In the MAML inner loop, for each task, conditioning on its S_train^aug (S_train^G, or S_train^*), we adapt the parameters of h^cls (h^G, or h^real) using few gradient updates. For each task, the adapted h^cls (h^G, or h^real) is evaluated on the corresponding S_test. In the MAML outer loop, we average the classification objective on S_test across the batch of tasks. In a similar way, we compute the collaborative objective in the pre-trained feature space. The final loss is used to update the initial MAML model and the hallucinator.
>
> 2. Table 4 … the results of PMN w/G:
>
> We have included the result of PMN w/ G (Wang et al, 2018) in Table 4, and updated the submission accordingly. Our approach significantly outperforms Wang et al., 2018. Here, we summarize the comparison:
>
> Method                                                  n=1                  n=5
> PMN w/ G (Wang et al, 2018)     62.28±0.53     78.28±0.62
> PMN w/ G + PECAN (Ours)         63.93±0.40     80.58±0.29
>
> 3. … hallucinating examples for regression tasks:
>
> In our paper, we mainly focus on few-shot classification tasks. However, our approach is general and applies to few-shot regression tasks as well. When extending our approach to address regression tasks, we need to slightly modify the design of the hallucinator G. Specifically, in classification tasks, the hallucinator G takes as input a seed example (x, y) and outputs a hallucinated example (x’, y’)=(G (x, z), y), where the class label y’=y, indicating that the hallucinated example belongs to the same class as the seed example. In contrast, in regression tasks, y becomes a continuous quantity, thus we cannot enforce the constraint y’=y. To address this issue, we modify the hallucinator G to directly generate the tuple (x’, y’) = G (x, y, z). This is achieved by the concatenation of x and y as input to G.
>
> We incorporated our hallucinator with MAML and evaluate on the sinusoidal regression task proposed in (Finn et al., 2017). Our ‘MAML w/ G + PECAN’ outperforms MAML and ‘MAML w/ G’ for the regression task as well. Here we briefly summarize the mean squared error comparison:
>
> Method                                               n=5
> MAML (Finn et al., 2017)                 0.84
> MAML w/ G (Wang et al, 2018)      0.67
> MAML w/ G + PECAN (Ours)          0.52
>
> We have revised the submission accordingly to emphasize classification tasks, especially in the summary of contributions in Section 1, as well as showing our generality to regression tasks in the appendix A.7.

---

### Public Comment · ~Temporary_Temp1 · 2019-10-24
**Clarification on constraint**

Thanks for the work, and congratulations on achieving such good results!

I have concerns regarding the objective function.

As far as I understand, you are imposing that Performance(Query | Augmented data) = Performance(Query | Support). If this is the case, then a trivial solution would be when the augmented data = support data (i.e. the hallucinator is essentially an autoencoder).

Am I missing something? Would you have some visualisations of the hallucinated images?

---

> ### Author Response · Authors · 2019-11-15
> **Thank you for the comments and interest in our approach.**
>
> 1. … imposing that Performance(Query | Augmented data) = Performance(Query | Support):
>
> We are not imposing that Performance(Query | Augmented data) = Performance(Query | Support). Instead, we target Performance(Query | Augmented data) = Performance(Query | Larger set of real data). This is explained in the paragraph “Soft precision-inducing hallucinator” in Section 4, page 5. Specifically, given an initial large training set S^*_{train}, which contains n^∗ examples for each of the m classes, we randomly sample n (n << n^∗ ) examples per class, and obtain a subset S_{train}. From S_{train}, the hallucinator G generates n^∗ examples per class as S^G_{train}. This produces two training sets: S^*_{train} with real examples and S^G_{train} with hallucinated examples, where both contain the same number of examples. Importantly, note that S^G_{train} is hallucinated from the subset S_{train} instead of the initial large set S^*_{train}, and because n << n^∗ , we rule out the trivial hallucinator.
>
> 2. Would you have some visualizations of the hallucinated images:
>
> Our hallucination is performed in the pre-trained feature space, and visualizing them directly is not intuitive. Instead, we provided a t-SNE plot of hallucinated samples in Figure 3. In the appendix A.8, we have included an additional visualization of hallucinated examples in the pixel space, using the nearest neighbor real image in the feature space, corresponding to each hallucinated feature sample.

---

### Author Response · Authors · 2019-11-15
**General Response**

We thank the reviewers for their positive comments --- our “novel and interesting idea” (R3) to few-shot learning achieves “consistent improvement in performance versus various state-of-the-art meta-learning algorithms’’ (R1) with “many experiments” (R1) and “quite nice and thorough ablation studies” (R1). We address all the comments below and have also updated the submission accordingly.

In the revised submission, we: (1) emphasized few-shot classification and included few-shot regression results in the appendix A.7, (2) included additional comparisons suggested by the reviewers, and (3) included visualization of the hallucinated examples suggested by the public comment in the appendix A.8.

---

### Decision · Program_Chairs · 2019-12-19

**Decision:**

Reject

**Comment:**

This paper describes a new approach to meta-learning with generating new useful examples.

The reviewers liked the paper but overall felt that the paper is not ready for publication as it stands.

Rejection is recommended.